DATA RELEASE

# Genome assembly of the bearded iris, *Iris pallida* Lam.

Robert E. Bruccoleri[1], Edward J. Oakeley[2], Ann Marie E. Faust[3],
Marc Altorfer[2], Sophie Dessus-Babus[2], David Burckhardt[2], Mevion Oertli[2],
Ulrike Naumann[2], Frank Petersen[2] and Joanne Wong[2,*]

1 Congenomics, LLC, Glastonbury, CT, USA
2 Novartis Institutes for BioMedical Research, Novartis Campus, 4056, Basel, Switzerland
3 Novartis Institutes for BioMedical Research, 250 Massachusetts Avenue, Cambridge, MA, USA

## ABSTRACT

Irises are perennial plants, representing a large genus with hundreds of species. While cultivated extensively for their ornamental value, commercial interest in irises lies in the secondary metabolites present in their rhizomes. The Dalmatian Iris (*Iris pallida* Lam.) is an ornamental plant that also produces secondary metabolites with potential value to the fragrance and pharmaceutical industries. In addition to providing base notes for the fragrance industry, iris tissues and extracts possess antioxidant, anti-inflammatory and immunomodulatory effects. However, study of these secondary metabolites has been hampered by a lack of genomic information, requiring difficult extraction and analysis techniques. Here, we report the genome sequence of *Iris pallida* Lam., generated with Pacific Bioscience long-read sequencing, resulting in a 10.04-Gbp assembly with a scaffold N50 of 14.34 Mbp and 91.8% complete BUSCOs. This reference genome will allow researchers to study the biosynthesis of these secondary metabolites in much greater detail, opening new avenues of investigation for drug discovery and fragrance formulations.

**Subjects** Genetics and Genomics, Botany, Plant Genetics

**Submitted:** 25 July 2023

\* Corresponding author. E-mail:
joanne.wong@novartis.com

Preprint submitted at https: //doi.org/10.1101/2023.08.29.555454

## BACKGROUND AND CONTEXT

The family *Iridaceae* comprises at least 250 known species and many hybrid cultivars. Traditionally grown for their ornamental value, irises are known to possess bioactive compounds in their tissues, particularly the rhizomes. Researchers have isolated PTP1B inhibitors from *Iris sanguine* Donn ex Hornem. (so named for Jens Wilken Hornemann) [1], antioxidant isoflavonoids [2] and the cytotoxic triterpenoid Belamchinenin A [3] from *Iris domestica*, the anti-*Helicobacter pylori* O-methylated flavonoid irigenin from *Iris confusa* Sealy [4], and anti-biofilm extracts from several iris species [5]. These compounds were extracted from their producer organisms, requiring several kilograms of plant material to yield compounds in milligram quantities [1, 3]. The bioactive compounds found in *Iris* species are secondary metabolites, produced through multi-enzyme biosynthetic cascades. Genetic engineering and synthetic biology allow researchers to reconstruct valuable biosynthetic pathways in other host organisms, providing the opportunity for large-scale fermentation and extraction; however, one requires access to the genes in such pathways to reconstruct them for large-scale expression and production of the secondary metabolites.

The Dalmatian iris *Iris pallida* Lam. (NCBI:txid29817, Figure 1, so named after Jean-Baptiste Lamarck) is a member of the *Iridaceae* family that produces bioactive

**Figure 1.** Representative specimen of *Iris pallida* (Photo credit: A.M.E.F.).

compounds. Its triterpenoid iridals act as ligands for the RasGRP family of diacylglycerol or phorbol ester receptors [6], and the triterpenoid iripallidal has been shown to inhibit AKT/mTOR and STAT3 signaling pathways in glioblastoma [7]. The closely related *Iris × germanica* L. also produces triterpenoids with anti-proliferative activity [8]. Given the wealth of bioactive compounds produced by iris species, we sought to fully sequence the genome of *Iris pallida* Lam. Karyotype analysis of *Iris pallida* Lam. revealed a diploid organism with 12 unique chromosomes [9].

Two *Iris* genomes have been published: *Iris sibirica* L. and *Iris virginica* L. (so named after Carl Linnaeus) [10]. While the approach involved short-read sequencing (Illumina 2 × 150 base pairs [bp] paired-end, followed by Spades assembly), longer read platforms such as Pacific Biosciences (PacBio) and Bionano Genomics are better suited to bridge across repetitive sequences, which account for a substantial portion of eukaryotic genomes and are expected to be common in the case of *Iris* species. Besides this, the chloroplast genome from *Iris speculatrix* Hance was sequenced to understand the phylogeny of the species [11], a large-scale RNA-seq transcriptional profile was generated for *Iris japonica* to investigate winter dormancy patterns [12], and a transcriptomic profiling effort was undertaken in *Iris × germanica* L. to understand reblooming mechanisms [13]. The next closest relative to the iris whose genome has been fully sequenced is *Asparagus officinalis* L. [14]. Using PacBio long-read sequencing technology, we obtained a full genomic sequence of 10.04 Gbp for *Iris pallida* Lam. from leaf tissue. Estimates of other *Iris* species' genome sizes range from 2 to 30 Gbp [15]; this range is in line with the genome size of *Iris pallida* Lam. in this study. From RNA extracted from rhizome and leaf tissues, we again

used PacBio sequencing technology to obtain an RNA transcriptional profile of *Iris pallida* Lam. The genome annotation was completed with PacBio transcripts, and all abundance numbers were obtained from PacBio data.

The genomic sequence and transcript information of *Iris pallida* Lam. will allow researchers to identify enzymes responsible for bioactive compounds, improving understanding of the biosynthetic pathways that generate bioactive compounds in the plant. This genome sequence and transcript data will also allow researchers to understand phylogenetic relationships between irises and other plant species and facilitate DNA and RNA sequencing efforts for other *Iris* species.

## METHODS

### Genome assembly sample and sequencing

For genomic DNA extraction, four 50-mL Falcon centrifuge tubes were each filled with 10 mL of extraction buffer (2% CTAB, 1.4 M NaCl, 20 mM ethylenediaminetetraacetic acid [EDTA], 100 mM Tris-HCl at pH 8.0, and 0.2% beta-mercaptoethanol. The tubes were warmed to 60 °C in a water bath. Apre-cooled mortar was filled with liquid nitrogen to a depth of 3 cm. Sterile sand was added to a depth of 0.5 cm. Five young iris leaves, approximately 20 cm in length, were cut from the plant and immediately cut into 2-cm lengths and submerged in liquid nitrogen. They were then ground into a fine powder, with additional liquid nitrogen carefully added as needed. Approximately 25% of the iris–sand powder was added to each warm extraction buffer tube and mixed by inversion 3 to 4 times. The mixture was incubated at 60 °C for 30 min and mixed by inversion every 5 to 10 min. Then, 10 mL of phenol:chloroform:isoamyl alcohol at a ratio of 25:24:1, was added and gently mixed by continuous inversion for 1 min. The sample was centrifuged at 6000 × g for 10 min to separate the phases. The upper phase was carefully removed and transferred to a 50-mL phase-lock gel tube (Eppendorf), and 10 mL of chloroform:isoamyl alcohol at a ratio of 24:1, was added and gently mixed by inversion for 1 min. The sample was centrifuged at 6000 × g for 10 min, after which time the aqueous phase above the phase-lock wax was removed and transferred to a fresh tube. An equal volume of isopropanol was added to the tube, and the sample was mixed by inversion until a gelatinous mixture of nucleic acids formed. This nucleic acid mixture was removed with a glass rod and washed three times with ice-cold 70% ethanol. The nucleic acid was allowed to air dry and then dissolved overnight in Tris–EDTA buffer. The sample was assigned the Novartis tracking ID AS_SAM_17_03QT. Library preparation used continuous long-read methods for genomic DNA sequencing for the Sequel 1 instrument, as per the manufacturer (PacBio)'s instructions. For optical mapping, a frozen sample of the *Iris pallida* Lam. leaf (GSM-AAB282) was shipped on dry ice to Bionano Genomics (San Diego, CA, USA), which sequenced the leaf genome to generate a genomic optical map.

A summary of sequencing data for this study is listed in Table 1. A total of 236 single-molecule real-time (SMRT) cells were used to produce 824,109,057,251 bp of genomic sequence data. The average length and N50 values for the PacBio subreads were 7,422 and 17,373 bp, respectively.

### Transcriptome samples and sequencing

Iris tissue samples (leaf and rhizome) were ground in liquid nitrogen, and RNA was extracted using the RNeasy Plant Mini Kit (Sigma Aldrich) and QIAshredder spin columns

**Table 1.** Summary of sequencing data generated in genome assembly of *Iris pallida* Lam.

| Sequencing platform | Data type | Tissue used | Raw data (bp) | Accession |
|---|---|---|---|---|
| Pacific Biosciences | RNA sequencing | Leaf | 45,352,970,323 | GKDR00000000 |
| Pacific Biosciences | RNA sequencing | Rhizome | 74,815,065,674 | GKDS00000000 |
| Pacific Biosciences | DNA sequencing | Leaf | 824,109,057,251 | JANAVB010000000 |

(Qiagen). The RNA samples were then treated with the TURBO DNA-free Kit (Thermo Fisher Scientific). Library preparation was carried out according to the procedures for Isoform Sequencing (Iso-Seq) using the Clontech SMARTer PCR cDNA Synthesis Kit with a BluePippin DNA Size Selection system. The rhizome sample of *Iris pallida* Lam. was sequenced using 10 SMRT cells and generated 74,815,065,674 bp of subread sequences, while the leaf sample was sequenced using 11 SMRT cells and generated 45,352,970,323 bp. Each of these datasets was processed using the PacBio Circular Consensus Sequencing (CCS) algorithm v5.1.0 (RRID:SCR_021174) of SMRT Link using the ccs2 pipeline named sa3_ds_ccs.

## Genome assembly and annotation

Falcon_unzip [16] was used to assemble the PacBio long read dataset. We used the Conda channels defaults, bioconda, and conda-forge to install pb-assembly, pbmm2, and genomicconsensus, as of July 24, 2019. Falcon-kit v1.4.2 (RRID:SCR_016089) and falcon-unzip v1.3.3 modules were present in our execution of Falcon_unzip. Falcon_unzip produced two assembly files – the primary contigs and the haplotigs – which are contigs representing variant genomic sequences that are similar but not identical to the primary contigs.

A second genomic DNA sample was shipped to Bionano Genomics to generate genomic optical maps. The resulting optical maps were used to scaffold genome assembly using the HybridScaffolding pipeline in the Bionano Genomics Solve package v3.2.1_04122018. Then, the PacBio Arrow algorithm implemented in the sl_resequencing2 pipeline present in SMRT-Link-7.0.1 was used to polish the Bionano Genomics hybrid assembly using original PacBio reads. The total elapsed time was 309 h (final output folder named arrow_iris_20191208). Telomeres were predicted using the FindTelomeres Python script from Jana Sperschneider [17] (RRID:SCR_024403).

## Transcriptome assembly and annotation

The lima program from PacBio v.1.6.1 (commit v1.6.1-1-g77bd658) was run to identify all CCS sequences with the expected 3′ and 5′ sequences. Then, the IsoSeq3 program from PacBio (commit v0.4.0-121-g22a3096*; (RRID:SCR_022749) [18] was used with the cluster option to group the CCS sequences into transcripts. The IsoSeq3 polish option was used to improve transcript accuracy. Prior to the availability of IsoSeq3 software, the PacBio RNA sequence data were analyzed using tools from IsoSeq1 and IsoSeq2 software distributions from PacBio, but the analysis was limited by the slow run time of the earlier algorithms. We also included any additional transcripts from these older algorithms in our genome annotation if they improved matches to the genome sequence compared with IsoSeq3 transcripts. Using the transcripts, the NCBI BLASTN algorithm v2.2.8 (RRID:SCR_001598) was used against the Bionano Genomics-scaffolded genome to identify probable locations for the corresponding genomic DNA. Then, Exonerate v2.2.0 (RRID:SCR_016088) [19] with the cdna2genome model was used to find the likely genomic location as well as putative exons and introns for each transcript. A maximum intron size of 30,000 bp was used



**Table 2.** Statistics for the *Iris pallida* Lam. genome.

| Date | Total length (bp) | N50 (bp) | Longest contig (bp) | Number of contigs | Coverage | Comments |
|---|---|---|---|---|---|---|
| 28-Sep-2019 | 10,460,090,820 | 583,967 | 4,430,189 | 38,684 | 78.8 | Falcon_unzip, primary contigs |
| 28-Sep-2019 | 2,642,332,941 | 101,281 | 1,374,365 | 38,684 | N/A | Falcon_unzip, haplotigs |
| 24-Dec-2019 | 13,489,134,452 | 14,342,615 | 85,218,729 | 45,374 | 61.1 | Bionano Genomics plus Arrow |

initially for the search, with a minimum match percentage of 80%. For all transcripts for which no genomic location was found, a second Exonerate run was attempted, with a maximum intron size of 2,000,000 bp. This two-stage process was used to reduce the total computer run time required to search for all transcripts.

Finally, all predicted exon and intron locations were loaded into a local relational database to facilitate the preparation of a genome submission to the National Center for Biotechnology Information (NCBI), which included coding region predictions based on the transcript RNA sequences.

When preparing the genome submission to NCBI, we aimed to identify likely gene products. We used the BLASTX (RRID:SCR_001653) [20] algorithm to find reasonable matches of the translated transcriptome sequences against three publicly available plant proteomes: *Asparagus officinalis* L., *Oryza sativa* L. Japonica Group, and *Zea mays* L., with an E-score threshold of 0.001.

Because many transcripts aligned to overlapping regions of the genome, we reported only one transcript per region; we were unable to rank multiple aligning transcripts. We used the Exonerate alignments that spanned no more than 1 million base pairs, because a spot check of the very large alignments appeared to be artifactual and obscured smaller groups of alignments. For each region, we chose the transcript whose genomic sequence yielded the longest open reading frame based on the standard genetic code. We reported this coding sequence in the NCBI submission along with the product name from the BLASTX search, and we included a note providing the *E*-value and bit-score of the BLASTX alignment along with the RefSeq identifier of the plant protein.

## RESULTS AND DISCUSSION

### Genome

It is challenging to generate accurate, relatively complete plant genome assemblies due to their large size, heterozygosity, and high frequency of repeat sequences. For these reasons, PacBio long-read sequencing technology was used to generate the *Iris pallida* Lam. genome assembly. The total size of the PacBio genome assembly was 10.46 Gbp. To enhance the assembly, we used Bionano Genomics optical mapping, since optical mapping on top of long-read sequencing is beneficial for producing higher quality plant genome assemblies [21, 22]. The total size of the genome assembly after Bionano Genomics scaffolding was 13.49 Gbp (Table 2). The additional size of the scaffolded genome was due to differing haplotigs in the phased assembly from Falcon Unzip.

Because primary contigs and haplotigs were included in the scaffolding process, many regions of this genome sequence are near duplicated. These near duplications are important to the future analysis of this genome because we cannot determine which allele of a heterozygous gene is functional.

Benchmarking Universal Single-Copy Orthologs (BUSCO) v 5.4.7 (BUSCO, RRID:SCR_015008) [23, 24] was run on the Bionano Genomics scaffolded assembly (Table 3)

**Table 3.** Completeness of the *Iris pallida* Lam. genome as evaluated by BUSCO.

| Count | Percentage of searched BUSCOs | Description |
|---|---|---|
| 234 | 91.8 | Complete BUSCOs (C) |
| 40 | 15.7 | Complete and single-copy BUSCOs (S) |
| 194 | 76.1 | Complete and duplicated BUSCOs (D) |
| 2 | 0.8 | Fragmented BUSCOs (F) |
| 19 | 7.4 | Missing BUSCOs (M) |
| 255 | 100.0 | Total BUSCO groups searched |

**Table 4.** Transcriptome statistics for *Iris pallida* Lam.

| PacBio subreads by tissue | Total transcripts | Average length (bp) | N50 (bp) |
|---|---|---|---|
| Leaf | 96,680 | 1,472 | 1,654 |
| Rhizome | 140,135 | 1,641 | 1,759 |

using both Augustus gene modeling software and the maize species parameters provided in Augustus. The lineage dataset was eukaryota_odb10, and the BUSCO mode was set to euk_genome_aug. The GC content of the genome was 41.2%. Compared with other plant genomes, the completeness of our assembly is reasonable with regards to genome–transcript alignment and BUSCO scores [25]. A small number of sequences were omitted from publication by NCBI due to short size or discovery of vector sequence contamination. Thus, the number of scaffolds for this genome reported by NCBI is slightly smaller than the number reported in Table 2.

## Transcriptome

Rhizome and leaf tissue samples were processed for RNA sequencing data. Statistics are shown in Table 4. For the rhizome sample, 3,032,725 CCS sequences were produced after processing by lima, resulting in 133,484 high-quality transcripts and 6,959 low-quality transcripts. For the leaf sample, 1,910,385 CCS sequences were produced after lima processing, resulting in 91,528 high-quality transcripts and 5,156 low-quality transcripts. Both high- and low-quality transcripts were used in the annotations. The CCS reads after lima processing were deposited into the NCBI Short Read Archive. There were 96,680 transcripts reported for the leaf sample and 140,135 transcripts reported for the rhizome sample. We found a total of 63,944 transcript-identified coding regions.

Out of 236,815 transcripts determined by the PacBio Isoseq3 method, 212,672 were aligned to the genome using Nucleotide BLAST, an alignment percentage of 89.8%. All transcripts that aligned using BLAST were then realigned against the *Iris pallida* Lam. genome in the vicinity of the genome matched by BLAST. The percentage of successful Exonerate alignments was 88.1%. The quality of alignment of the transcriptome to the genome was very high.

Exonerate computes the number of identical base matches for its alignments, as well as the number of mismatches. For all Exonerate alignments, the ratio of identical base matches to the sum of base matches and mismatches was 98.1%. Given that the individual plant used for RNA isolation was different from the individual plant used for genomic DNA isolation, this result represents an excellent agreement of nucleotide sequences.

After submitting the *Iris pallida* Lam. transcript sequences to NCBI, NCBI reported that 230 rhizome sample transcripts contained sequences from species other than *Iris pallida* Lam. Most of these contaminant sequences were from fungal species, an expected result given that the rhizome tissue sample was removed from soil. These sequences were removed from the final submitted transcriptome and were not used in the annotation of the *Iris pallida* Lam. genome that was submitted to NCBI.

### Telomeres

In the Bionano Genomics scaffolded assembly, a total of 26 scaffolds had telomere sequences at their ends. These telomere data were included in the NCBI submission. We did not identify any contigs or scaffolds that had telomeres at both ends.

With 12 unique chromosomes [9], *Iris pallida* Lam. would be expected to have 24 unique concatenations of telomeres with chromosome end sequences. Given the inclusion of haplotigs into the genome assembly as well as its draft quality, the identification of 26 telomeres in our assembly is consistent with the observed chromosome number.

### DATA VALIDATION AND QUALITY CONTROL

The genome quality was assessed first by BUSCO analysis. We found 91.8% complete BUSCOs, of which 15.7% were complete and single-copy. Second, we assessed quality by mapping RNA transcripts to the genome assembly. Here, we found an 88.1% success rate for Exonerate alignments. Thus, the DNA and RNA data are in strong agreement, indicating high-quality sample collection, data generation, data processing, and data analysis for the *Iris pallida* Lam. genome assembly.

The leaf and rhizome iris samples were collected from a private garden in Basel, Switzerland. Tissue samples were collected from the same plant specimen. Their utilization in our research was in full compliance with the Nagoya Protocol [26].

### REUSE POTENTIAL

Plant genomes and transcriptomes are essential for understanding the secondary metabolite (also known as natural product) biosynthetic pathways that produce these valuable molecules. Natural products are often extracted from producer species without knowledge of their biosynthesis, so industrial-scale production of natural products is hampered by plant availability. Iris species produce natural products in many compound classes [1–5] but, until now, no iris genome has been sequenced using PacBio long-read sequencing. The long-read genome assembly and mapped transcriptome of *Iris pallida* Lam. will allow researchers to sequence parts, or complete genomes, of other iris species – broadening our understanding of those natural products that are common to *Iris,* and those that are species-specific and responsible for the unique aromas and biological properties of irises. Additionally, identification of *Iris* genes and pathways might aid researchers who study the phylogenetic relationships of plant families.

### DATA AVAILABILITY

The genome assembly is available at NCBI under accession JANAVB010000000. The BioProject identifier at NCBI is PRJNA813844. The BioSample accessions are available within the above BioProject description at NCBI. The Iso-Seq transcript data from the rhizomes and leaves are available at NCBI in the Transcriptome Shotgun Assembly (TSA)

Database under accessions GKDS00000000 and GKDR00000000, respectively. The CCS reads that were used as input files to the Iso-Seq algorithm are available in the NCBI Short Read Archive (SRA). The SRA accession for the rhizome transcript reads is SRR22228979, and for the leaf transcript reads, SRR22228019. In addition, all of the subreads from the genome sequencing are available in the SRA under the above BioProject identifier. Additional data is available in GigaDB [27].

## LIST OF ABBREVIATIONS

bp, base pairs; BUSCO, Benchmarking Universal Single-Copy Orthologs; CCS, circular consensus sequencing; EDTA, ethylenediaminetetraacetic acid; Hornem., Jens Wilken Hornemann; L., Carl Linnaeus; Lam., Jean-Baptiste Lamarck; NCBI, National Center for Biotechnology Information; PacBio, Pacific Biosciences; SMRT, single-molecule real-time; SRA, Short Read Archive.

## DECLARATIONS

### Ethics approval

The authors declare that ethical approval was not required for this type of research.

### Competing Interests

REB is a paid consultant to the Novartis Institutes for BioMedical Research, Inc. All other authors declare no competing interests.

### Authors' contributions

REB, AMEF, EJO, UN, FP, and JW contributed to the study design. AMEF, EJO, MA, and JW collected and processed iris tissue samples for nucleic acid extraction. DB, MA, SDB, and MO prepared the nucleic acid libraries and performed the sequencing. REB and EJO analyzed genomic and transcriptomic data, including assembly, scaffolding, and polishing the genome. REB processed, formatted, and submitted genomic and transcriptomic data to NCBI, while REB, AMEF, and JW wrote and revised the manuscript.

### Funding

This work was funded by the Novartis Institutes for BioMedical Research, Inc.

### Acknowledgements

The authors would like to acknowledge Kerstin Oelkers for assisting with the sequencing library preparation; Jasmin Hägele for assisting in the watering of the iris plants; Tim Schuhmann for providing analytical support; Brigitta Liechty for providing an *Iris pallida* Lam. plant specimen; and Maulik Thaker and Horst Hemmerle for contributions to alternate genomic sequencing efforts.

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
