## [Editor Report]

Editor’s AssessmentIrises on top of being a popular and beautiful ornamental plant, have wider commercial interest due to the many interesting secondary metabolites present in their rhizomes that have value to the fragrance and pharmaceutical industries. Many of these have large and difficult to assemble genomes, and to fill that gap the Dalmatian Iris (Iris pallida Lam.) is sequenced here. Using PacBio long-read sequencing and bionano optical mapping to produce a giant 10Gbp assembly with a scaffold N50 of 14.34 Mbp. The authors didn’t manage to handle the haplotigs separately or to study the ploidy, but as all of the data is available for reuse others can explore these questions further. This reference genome should also allow researchers to study the biosynthesis of these secondary metabolites in much greater detail, opening new avenues of investigation for drug discovery and fragrance formulations.

---

## [Reviewer Report]

Reviewer name and names of any other individual's who aided in reviewer Kang ZhangDo you understand and agree to our policy of having open and named reviews, and having your review included with the published papers. (If no, please inform the editor that you cannot review this manuscript.)YesIs the language of sufficient quality?YesPlease add additional comments on language quality to clarify if needed
I found several sentences confusing: P2L8 (Is the DNA/RNA extraction particularly difficult for iries?), and P9L5 (wording).Are all data available and do they match the descriptions in the paper? YesAdditional CommentsAre the data and metadata consistent with relevant minimum information or reporting standards? See GigaDB checklists for examples <a href="http://gigadb.org/site/guide" target="_blank">http://gigadb.org/site/guide</a>YesAdditional CommentsIs the data acquisition clear, complete and methodologically sound?YesAdditional CommentsIs there sufficient detail in the methods and data-processing steps to allow reproduction?YesAdditional CommentsIs there sufficient data validation and statistical analyses of data quality? YesAdditional CommentsIs the validation suitable for this type of data?YesAdditional CommentsIs there sufficient information for others to reuse this dataset or integrate it with other data?YesAdditional Comments1. P7L20. The basic stats of the subreads should be introduced before the assembling process. 2. The authors should provide more methodological details about the BUSCO assessment, such as the database version, the mode (genome or protein), etc. 3. I am curious about the genome size enlargement introduced by the scaffolding. Were different haplotigs (from different haplotypes) were used for scaffolding, and why? I suppose that only the primary haplotigs should be used. 4. Considering the high proportion of duplicated BUSCO genes, I wonder whether the iris sequenced is a polyploid or not? Please clarify it in the Background.Any Additional Overall Comments to the AuthorDr. Wong and his colleagues reported a genome assembly of iris using the PacBio technology. Due to the huge genome size, the generated data volume is impressive. Although the quality of the assembly is not so satisfying, it is reasonable considering the genome size and the high heterozygosity, which is commonly found in many flowers. Overall, the methods used in this work are well described, and the data could be accessed. I only get several minor points regarding the details during the assembling process.RecommendationMinor Revision

---

## [Reviewer Report]

Upload additional filesDRR-202307-03/form/review.docxReviewer name and names of any other individual's who aided in reviewer Baocai HanDo you understand and agree to our policy of having open and named reviews, and having your review included with the published papers. (If no, please inform the editor that you cannot review this manuscript.)YesIs the language of sufficient quality?YesPlease add additional comments on language quality to clarify if needed
Are all data available and do they match the descriptions in the paper? YesAdditional CommentsAre the data and metadata consistent with relevant minimum information or reporting standards? See GigaDB checklists for examples <a href="http://gigadb.org/site/guide" target="_blank">http://gigadb.org/site/guide</a>YesAdditional CommentsIs the data acquisition clear, complete and methodologically sound?YesAdditional CommentsIs there sufficient detail in the methods and data-processing steps to allow reproduction?YesAdditional CommentsIs there sufficient data validation and statistical analyses of data quality? YesAdditional CommentsIs the validation suitable for this type of data?YesAdditional CommentsIs there sufficient information for others to reuse this dataset or integrate it with other data?YesAdditional CommentsAny Additional Overall Comments to the AuthorRecommendationMinor Revision